# Direct evidence that twisted flux tube emergence creates solar active regions

D. MacTaggart [1✉], C. Prior[2], B. Raphaldini[2], P. Romano [3] & S. L. Guglielmino [3]

The magnetic nature of the formation of solar active regions lies at the heart of understanding solar activity and, in particular, solar eruptions. A widespread model, used in many theoretical studies, simulations and the interpretation of observations, is that the basic structure of an active region is created by the emergence of a large tube of pre-twisted magnetic field. Despite plausible reasons and the availability of various proxies suggesting the accuracy of this model, there has not yet been a methodology that can clearly and directly identify the emergence of large pre-twisted magnetic flux tubes. Here, we present a clear signature of the emergence of pre-twisted magnetic flux tubes by investigating a robust topological quantity, called magnetic winding, in solar observations. This quantity detects the emerging magnetic topology despite the significant deformation experienced by the emerging magnetic field. Magnetic winding complements existing measures, such as magnetic helicity, by providing distinct information about field line topology, thus allowing for the direct identification of emerging twisted magnetic flux tubes.

[1] School of Mathematics and Statistics, University of Glasgow, Glasgow G12 8QQ, UK. [2] Department of Mathematical Sciences, Durham University, Durham DH1 3LE, UK. [3] INAF-Osservatorio Astrofisico di Catania, Via S. Sofia 78, I-95123 Catania, Italy. ✉email: david.mactaggart@glasgow.ac.uk

Twisted flux tubes are prominent candidates for the progenitors of solar active regions[1–4]. Twist allows a flux tube to suffer less deformation in the convection zone compared to untwisted tubes, thus allowing it to survive and reach the photosphere to emerge[2,5,6]. Further, simulations of twisted flux tube emergence have reproduced signatures that can be found in observations[1,3]. This has led to observational proxies (such as sigmoidal field lines in the atmosphere[3,7] and magnetic tongue patterns in magnetograms[8,9]) that are indicative of twisted tube emergence. Although highly suggestive, such proxies cannot provide direct and conclusive evidence of the emergence of a twisted flux tube from the convection zone to the solar atmosphere. This is because their signatures can also be created by magnetic fields that are not pre-twisted flux tubes[10]. Further, the signatures of these proxies can also be created by photospheric motions deforming simple magnetic fields, e.g., shearing flows along the polarity inversion line of an active region leading to sigmoidal field lines[11,12]. Some studies[13–15], by focussing on photospheric motions deforming magnetic fields, have concluded that pre-twisted magnetic fields do emerge. However, the question of whether or not twisted magnetic flux tubes create active regions has remained open, despite these important results. This is largely due to the complex nature of flux emergence.

Twist in a magnetic field is a manifestation of magnetic topology, which describes the connectivity of field lines. A classical measure of magnetic topology is magnetic helicity[16]. This quantity describes the field line topology (in terms of Gauss linkage[16] or winding[17], depending on the precise application) weighted by magnetic flux. Further, the flux of magnetic helicity through the photosphere can be measured in solar observations[18,19], so this topological quantity can potentially indicate what kind of magnetic topology is emerging. Many works[18–30] have studied the injection of this quantity in active regions, but a clear-cut indication of an emerging magnetic field's magnetic topology from magnetic helicity flux has remained elusive. One reason for this is that magnetic helicity combines two distinct properties, field line topology and magnetic flux, into one quantity. This combination can result in confounding the interpretation of the helicity flux. For example, an emerging field with simple (weak) field line topology could have a very strong field strength. Thus, a strong input of helicity does not necessarily indicate that complex (strong) magnetic topology is emerging into the solar atmosphere. Similarly, a magnetic field with a highly twisted topology but weak flux, can result in a weak helicity input, and so also not give an accurate indication of the true nature of the field line topology emerging into the atmosphere.

Magnetic winding[17] is a renormalisation of magnetic helicity that removes the magnetic flux weighting, and thus provides a direct measure of magnetic topology. Despite its close connection to magnetic helicity, magnetic winding can behave very differently in an evolving magnetic field and, hence, provide distinct information. Further, magnetic winding flux can be calculated from observations just like the helicity flux (there also exist local versions that can be estimated in the solar atmosphere[31]). Magnetic winding flux has been studied in simulations of magnetic flux emergence, including twisted flux tubes and more complex magnetic topologies[32,33]. Over a wide range of physical parameters, the accumulation of winding (the time-integrated winding flux) during the initial emergence of a twisted flux tube produces a consistent signature: an initial increase in the magnitude of the winding input followed by a plateau. This signature indicates that the twisted tube (or, at least, a substantial part of it) passes through the photospheric boundary (where the flux calculation is performed) and then essentially remains above this plane afterwards (until perhaps much later times when the active

region begins to decay, but we are only interested in the initial stages of emergence here). Even the effects of convection, in simulations, which act to drag down parts of the emerged field and create a serpentine geometry, have little effect on the winding input signature, as the magnetic topology is still dominated by the twist in the tube that remains primarily above the photospheric boundary. By contrast, the net magnetic helicity input can change sign during the emergence of twisted flux ropes due to the strength of convective down-flows, leading to a potential misinterpretation of the magnetic field's structure, i.e. a failure to diagnose the true twisted nature of the emerging field[33]. This is not to say that the magnetic helicity flux produces an erroneous signal but, rather, that it provides information based on a combination of factors and not just the global magnetic topology. For this reason, previous observational studies of magnetic helicity input produce complex signatures. From this perspective, the combination of both magnetic winding and magnetic helicity can provide much more detailed information about the dynamics and topology of emerging magnetic fields.

In this work, we show that magnetic winding flux can be used to detect the emergence of pre-twisted magnetic fields in solar observations, as well as in simulations. We show that magnetic winding produces a clear and consistent signature for emerging flux tubes that is not found by studying the magnetic helicity flux. Combining the information given by magnetic winding with other proxies and measures (including magnetic helicity) provides a much more detailed picture of the topological structure and evolution of an emerging solar active region.

## Results

**Simulation example.** An example of the winding signature described above, from a magnetohydrodynamic simulation of the initial stages of the emergence of a twisted flux tube, is shown in Fig. 1.

Figure 1(a) shows the emerged flux tube, at $t=2500$s, which has been deformed significantly by convection and developed a serpentine field line geometry. Any clear visual resemblance to a twisted tube is lost and the emerged magnetic field has the form of a collection of magnetic arcades. Indeed, it is important to point out that when we refer to twisted tube emergence, we are not implying that a coherent twisted flux tube emerges bodily from the convection zone to the corona. Instead, the tube emerges into the photosphere and the bulk of the twist remains there, continually being deformed by convection. As the magnetic field expands into the atmosphere, it evolves into one or several shear arcades, depending on the complexity of the region and the subsequent magnetic reconfiguration[1–3]. Therefore, when we speak of realistic twisted flux tubes in the convection zone, we refer to compact (in a tubular-like domain) magnetic fields with a net value of non-zero helicity, i.e. an overall net twist. Perturbations due to convection, which add twist locally, as seen in Fig. 1(a), but add zero net helicity to the magnetic field do not affect the magnetic winding, which is clear from Fig. 1(b).

Figure 1(b) shows the accumulation of winding above the photospheric boundary. There are three profiles to consider: the total winding accumulation, the braiding accumulation and the emergence accumulation. The calculation of the magnetic winding depends on two components of the magnetic field line velocity $\mathbf{u}$ on the planar photospheric boundary: a velocity due to in-plane motion, $\mathbf{v}_{||}$, and a velocity due to the emergence of the magnetic field, $-v_z\mathbf{B}_{||}/B_z$, where $\mathbf{v} = (\mathbf{v}_{||}, v_z)$ is the plasma velocity field and $\mathbf{B} = (\mathbf{B}_{||}, B_z)$ is the magnetic field (parallel subscripts indicate being parallel to the photospheric plane). The braiding accumulation $L_{\text{braid}}$ describes the winding input due to only the in-plane velocity and measures the entanglement of the

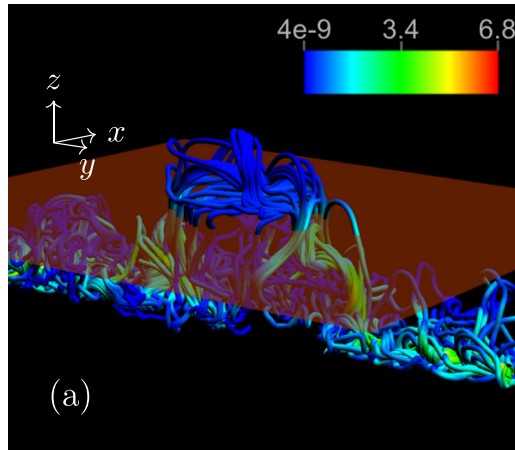

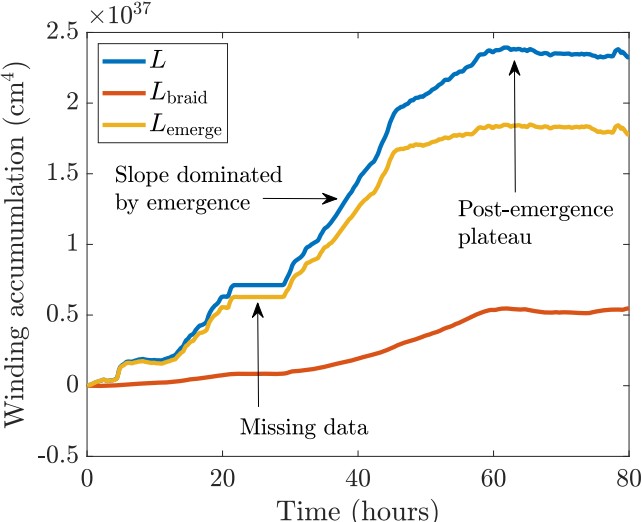

**Fig. 2 Winding accumulation for AR11318.** The emergence accumulation $L_{emerge}$ (yellow line), the braiding accumulation $L_{braid}$ (red line), and the total winding accumulation $L = L_{emerge} + L_{braid}$ (blue line) are displayed. The missing data are between $t = 20$ and $t = 30$ h. Source data are provided as a Source Data file.

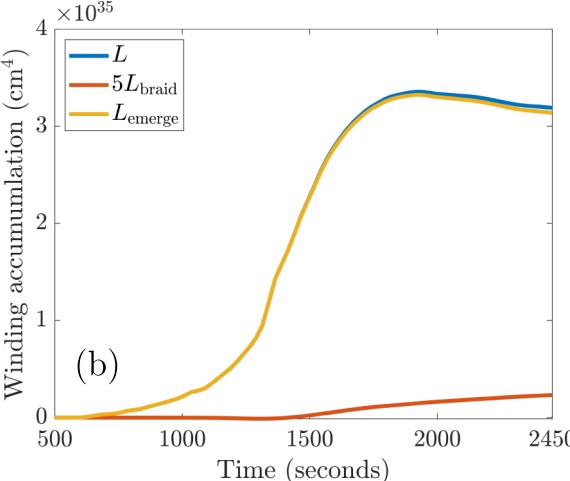

**Fig. 1 Simulation of the initial emergence of a twisted magnetic tube.** (**a**) displays field lines at $t = 2500$ s, with a slice indicating the horizontal photospheric boundary (red plane) where the magnetic field is measured. The $x$-, $y$- and $z$-directions are indicated. The colour bar indicates (dimensionless) magnetic field strength (to convert to physical units, see Methods: Numerical simulation details). Darker colours (mainly blue) indicate weaker field strengths and lighter colours (green and yellow) indicate stronger field strengths. In (**b**), the emergence accumulation $L_{emerge}$ (yellow line), the braiding accumulation $L_{braid}$ (red line), and the total winding accumulation $L = L_{emerge} + L_{braid}$ (blue line) are displayed. $5 \times L_{braid}$ is displayed on the figure in order to convey clearly how $L_{braid}$ develops in time.

magnetic field due to horizontal photospheric motions. The emergence accumulation $L_{emerge}$ describes the winding input due to only the emergence velocity and measures the contribution to winding due to pre-entangled emerging magnetic field (for more details, see Methods: Helicity and winding flux calculations). The total winding accumulation follows the signature described previously. Importantly, the emergence accumulation dominates strongly over the braiding accumulation. This result states that the input of magnetic winding into the atmosphere is due primarily to the emergence of a pre-entangled magnetic field with a net twist and not due to horizontal motions twisting the magnetic field at the photosphere. The signature in Fig. 1(b) indicates that the emerging entangled field is dominated by a positive twist of the magnetic field lines. More complex field line topologies with zero net twist have been shown not to create such a clear plateau accumulation signature[32]. The magnetic winding is robust enough to detect the twisted field line topology despite

the substantial deformation to the original flux tube in the convection zone. Another example of this, for an emerging twisted tube with perturbations adding zero net twist and no clear bipolar structure, is presented in the Supplementary Information.

**Observational analysis.** We now present direct evidence of twisted flux tube emergence in solar observations. For the purpose of this article, we focus on clean cases (isolated and coherent bipolar regions) that can be compared to, as closely as possible, the results from simulations. The active regions in the observations we consider are much larger than that of the simulation presented earlier. However, the result that we are testing for is not strictly scale-dependent, i.e., the same signature applies to large or small tubes, as long as they emerge above the photospheric boundary.

We first consider the National Oceanic and Atmospheric Region (NOAA) active region AR11318. We choose to study this region as it represents a small and simple bipolar region that can be compared to, within reason, flux emergence simulations. This region has also been studied in detail by measuring a range of different observational signatures[29]. These signatures, however, are still not enough to confirm that a coherent twisted flux tube has emerged to create this bipolar region. Figure 2 shows the winding accumulation for the initial emergence of AR11318, starting from 20:00UT on October 11th 2011, when the active region was located at $-15°$ with respect to the central meridian, and lasting for 80 h. This time span covers both the initial increase of line-of-sight magnetic flux and a period of its stabilisation[29]. To perform winding flux calculations, we use Space-Weather Helioseismic and Magnetic Imager (HMI) Active Region Patches (SHARP)[34] vector magnetograms taken by HMI on board of the Solar Dynamics Observatory (SDO), and we determine the plasma velocity using the Differential Affine Velocity Estimator for Vector Magnetograms (DAVE4VM)[35]. The SHARP vector magnetograms have a pixel resolution of 0".5 and a time cadence of 720 s.

Between $t = 20$ and $t = 30$ h, there are missing data, which result in an artificial plateau in the accumulation curves. Ignoring this data gap, the total winding accumulation $L$ (calculated with both velocity components) follows the signature of Fig. 1(b) (and

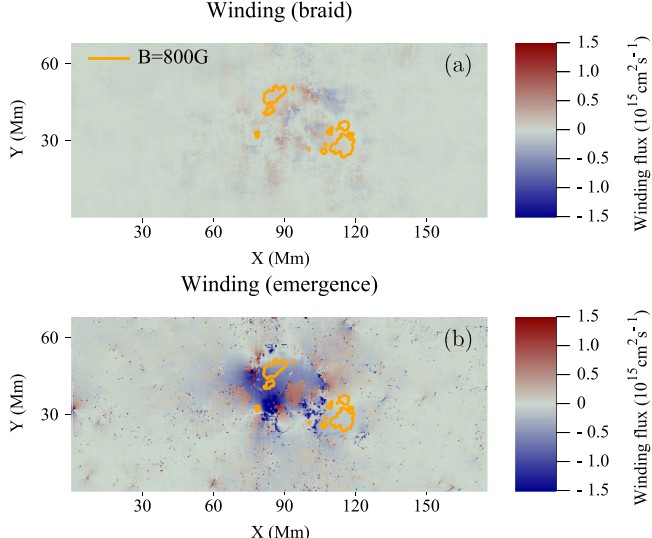

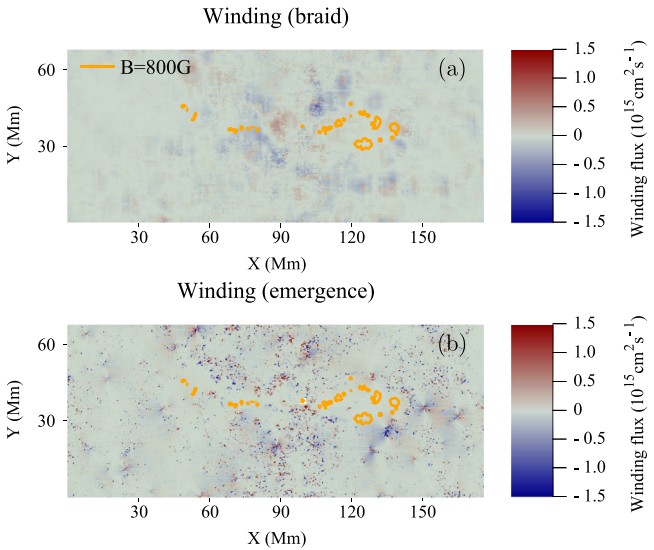

**Fig. 3 Winding flux distributions for AR11318 during the rise phase.** Distributions of (**a**) $dL_{braid}/dt$ and (**b**) $dL_{emerge}/dt$ of AR11318 at $t = 42$ h. The orange contours indicate a field strength of 800 G (and thus indicate the sunspot locations). X and Y are distances along the x- and y-directions.

**Fig. 4 Winding flux distributions of AR11318 during the plateau phase.** Distributions of (**a**) $dL_{braid}/dt$ and (**b**) $dL_{emerge}/dt$ of AR11318 at $t = 78$ h. The orange contours indicate a field strength of $B_z = 800$ G. X and Y are distances along the x- and y-directions.

other simulations[32,33]), namely a strong rise followed by a plateau. It is clear from Fig. 2 that the emergence accumulation dominates strongly over the braiding accumulation, and so the winding input is due primarily to the emergence of a pre-twisted structure rather than an untwisted structure whose twist develops in the solar atmosphere due to photospheric motions. If, during the data gap, the curves were to follow their approximately constant gradients (a reasonable assumption based on the behaviour of the gradients immediately before and after the data gap), the difference between $L_{emerge}$ and $L_{braid}$ would be even greater than that displayed in Fig. 2. This signature, together with those found previously[29] for AR11318 (magnetic tongues and the development of sigmoidal field lines) that follow the emergence evolution expected from simulations of twisted tube emergence[1–3], clearly and directly demonstrates that the region was formed by an emerging twisted flux tube.

Figure 3 displays distributions of (a) $dL_{braid}/dt$ and (b) $dL_{emerge}/dt$ of AR11318 at $t = 42$ h. The contribution due to emergence dominates the region between the sunspots, where the horizontal part of the twisted tube emerges. The contribution due to braiding is much weaker and spread throughout the domain. There are also coherent patches near the sunspots which move apart during the emergence of the region.

At later times, once the main bulk of the flux tube has emerged, and the winding accumulation reaches its plateau, the large-scale features of the magnetic winding distributions disperse. Figure 4 displays distributions of (a) $dL_{braid}/dt$ and (b) $dL_{emerge}/dt$ of AR11318 at $t = 78$ h.

The large-scale coherent features of the emergence contribution seen in Fig. 3(b) have all but disappeared as no significant magnetic field (i.e. on the active region scale) with nontrivial topology is emerging.

The behaviour of the winding profiles and distributions can be explained with the aid of the cartoon shown in Fig. 5. This cartoon indicates what parts of an emerging twisted magnetic field contribute to the winding accumulation. The emergence velocity is most prominent between the sunspots where $B_z$ is weaker and $\mathbf{B}_{\parallel}$ is at its strongest. It is between the sunspots where the main bulk of the twisted tube emerges and this results in a strong signal in the emergence accumulation (see Fig. 3(b)). The

in-plane velocity is stronger near the sunspots and is generally weaker between the sunspots. These considerations are important for understanding why the magnetic winding flux can detect the emergence of large-scale entanglement (twist in this case) clearly whereas the magnetic helicity flux may not. In magnetic helicity flux calculations, each point on the photosphere is weighted by $B_z$. The effect of this is to make the contribution of the emergence velocity small, since $B_z$ is weaker between the sunspots (for the simple emerging twisted tube under consideration here). Hence, the magnetic helicity flux input is dominated by the in-plane velocity contribution at the sunspots, which is weighted by strong $B_z$. Therefore, the magnetic helicity flux may not adequately diagnose the emergence of twisted magnetic field emerging between the sunspots. Figure 6 displays the helicity accumulation for AR11318. The importance of the emergence and braid components is now the opposite to that of the winding, in line with our description above. Figure 7 shows the helicity input distributions at $t = 42$ h. Again, as per the above discussion, the helicity input is dominated by the sunspots, due to the weighting of strong $B_z$, and so the bulk of the signature of the twisted flux tube emerging between the spots is missed.

It is of interest to note that the plateau of the magnetic winding accumulation (and, for this particular region, the plateau of the magnetic helicity accumulation) occurs after the plateau of the emerging line-of-sight magnetic flux accumulation (see Fig. 6 of Romano et al.[29]). Therefore, if increasing magnetic flux traditionally represents the emergence phase of the active region and the flux plateau represents the start of its decay phase, why do the winding and helicity accumulations continue to grow in the decay phase? The answer is that the magnetic flux accumulation does not provide a complete picture of the emergence process. Flux emergence simulations have long supported the two-stage picture of flux tube emergence, where a twisted tube can rise to the base of the photosphere and emerge into the atmosphere after a time delay[2,3,33]. The line-of-sight magnetic flux accumulation records the main flux build-up due to the sunspots, but does not take into account the emergence of the horizontal magnetic field between the sunspots. The magnetic winding and helicity fluxes do take this horizontal magnetic field into account, with the signal more dominant in the winding flux than the helicity flux.

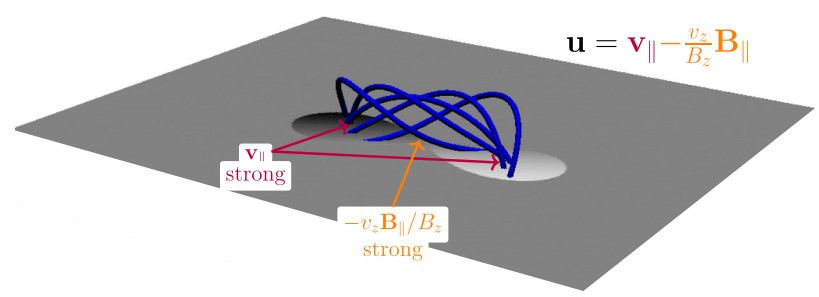

$$\mathbf{u} = \mathbf{v}_{\parallel} - \frac{v_z}{B_z}\mathbf{B}_{\parallel}$$

**Fig. 5 Influence of velocity components.** A cartoon showing field lines (in blue) of an emerging twisted flux tube above a planar boundary (shown in grey) representing the photosphere. A map of $B_z$ is displayed on the photospheric boundary. White indicates $B_z > 0$ and black indicates $B_z < 0$. The field line velocity $\mathbf{u}$ has two components: in-plane (purple) and emergence (orange), which play important roles at different parts of an emerging twisted flux tube, as indicated. Scale is not presented as the result applies to both large and small flux tubes.

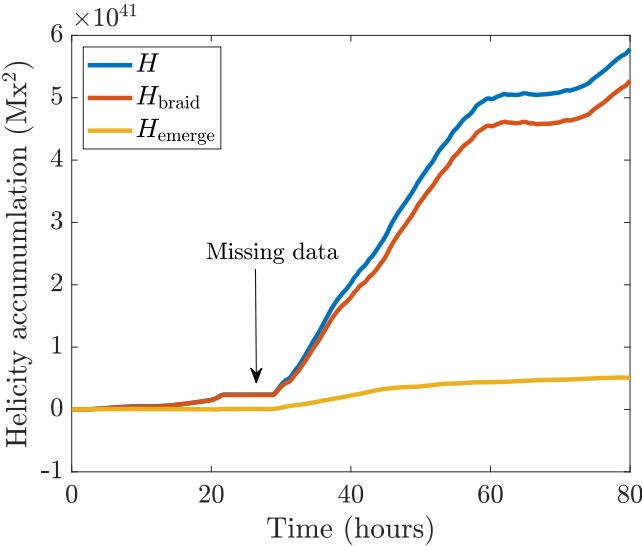

**Fig. 6 Helicity accumulation for AR11318.** The emergence accumulation $H_{\text{emerge}}$ (yellow line), the braiding accumulation $H_{\text{braid}}$ (red line), and the total helicity accumulation $H = H_{\text{emerge}} + H_{\text{braid}}$ (blue line) are displayed. Source data are provided as a Source Data file.

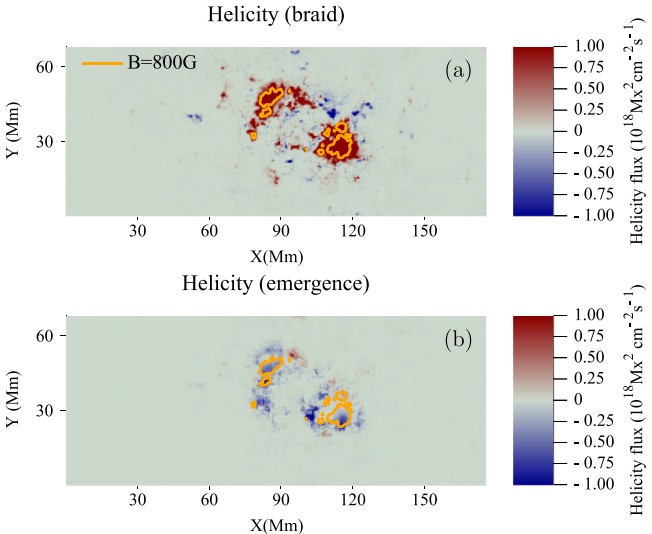

**Fig. 7 Helicity flux distributions.** Distributions of (**a**) $dH_{\text{emerge}}/dt$ and (**b**) $dH_{\text{braid}}/dt$ of AR11318 at $t = 42$ h. The orange contours indicate a field strength of $B_z = 800$G (and thus indicate the sunspot locations). X and Y are distances along the x- and y-directions.

Therefore, since the emergence of the horizontal part of the emerging magnetic field need not be co-temporal with the behaviour of the line-of-sight magnetic flux accumulation, the magnetic winding and helicity accumulations do not need to be in one-to-one time correspondence with the magnetic flux accumulation either. Also, the braiding of field lines at the photosphere is expected to continue after the initial phase of emergence due to subsequent photospheric motions, so this contribution can also result in the lack of a one-to-one time correspondence between the magnetic winding and helicity accumulations and the magnetic flux accumulation.

Other clean examples of twisted flux tube emergence can be found. Figure 8 shows the winding accumulation during the initial emergence of AR12203, starting at 13:00UT on 30th October 2014 and lasting for 80 h, just as for AR11318 in Fig. 2. In this region, a negative (left-handed) twist emerges into the atmosphere. Again, the emergence accumulation dominates strongly over the braiding accumulation and provides a clear signature that the magnetic topology of this region comes initially from the emergence and not photospheric shearing motions. Examining other signatures, such as those studied previously for AR11318[29], reveals a similar initial evolution. Figure 9 displays the helicity accumulation of AR12203. As for AR11318, the importance of the emergence and braiding

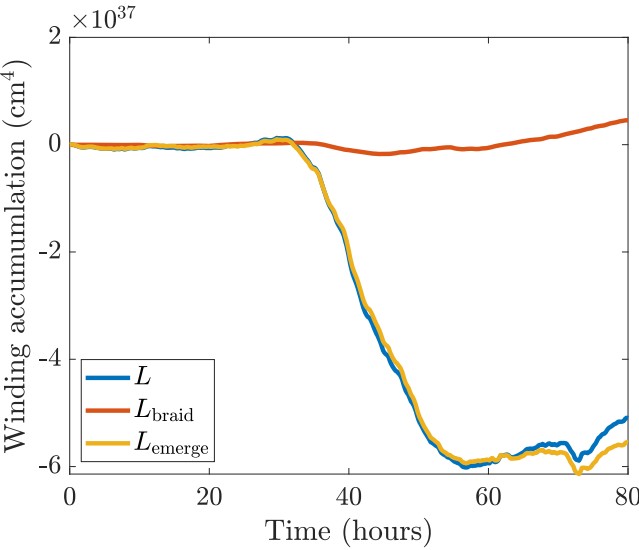

**Fig. 8 Winding accumulation of AR12203.** The emergence accumulation $L_{emerge}$ (yellow line), the braiding accumulation $L_{braid}$ (red line), and the total winding accumulation $L = L_{emerge} + L_{braid}$ (blue line) are displayed. Source data are provided as a Source Data file.

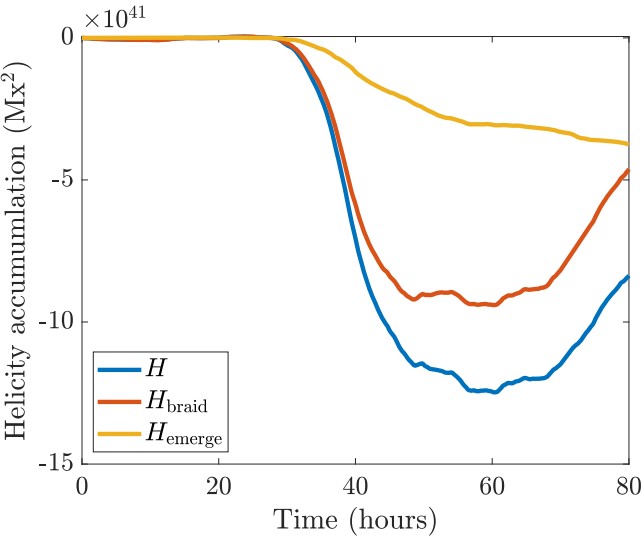

**Fig. 9 Helicity accumulation for AR12203.** The emergence accumulation $H_{emerge}$ (yellow line), the braiding accumulation $H_{braid}$ (red line), and the total helicity accumulation $H = H_{emerge} + H_{braid}$ (blue line) are displayed. Source data are provided as a Source Data file.

contributions has now switched compared to the winding accumulation. These helicity inputs, unlike the winding inputs, do not give a clear indication of how the topology of the emerging magnetic field enters into the atmosphere.

## Discussion

Magnetic winding, in combination with other measurable quantities (such as magnetic helicity) and proxies, provides a powerful analysis tool that can give direct information about magnetic topology. We have given examples of active region observations where the magnetic winding gives a clear indication that the emerging magnetic field is composed of pre-twisted magnetic field. This confirms that, as assumed in many theoretical studies, pre-twisted flux tubes play a fundamental role in active region formation. The pre-twisted magnetic field that emerges into the photosphere represents a source to explain, self-consistently, the shearing, rotational and compressional motions invoked in models of coronal mass ejection formation. These motions can develop due to the transportation of twist into the higher atmosphere as the magnetic field expands into the corona[1,3,36,37]. Although we have presented evidence that twisted flux tubes can create active regions, it is likely that other magnetic topologies also emerge to create active regions. This is an important area of research, for which magnetic winding will play a pivotal role.

## Methods

**Numerical simulation details.** The flux emergence simulation was performed using the open-source Lare3D code[38], which solves the equations of magnetohydrodynamics (MHD). The version we used includes an extended energy equation to allow for the modelling of convection, the details of which are described in a previous study[33]. The setup of the simulation is almost identical to that previous study but, for completeness, we describe the key details of the computational model. The nondimensional equations solved are those of compressible and ideal MHD,

$$\left(\frac{\partial}{\partial t} + \mathbf{v} \cdot \nabla\right)\rho = -\rho\nabla \cdot \mathbf{v}, \tag{1}$$

$$\rho\left(\frac{\partial}{\partial t} + \mathbf{v} \cdot \nabla\right)\mathbf{v} = -\nabla p + (\nabla \times \mathbf{B}) \times \mathbf{B} + \nabla \cdot \boldsymbol{\sigma} + \rho\mathbf{g}, \tag{2}$$

$$\rho\left(\frac{\partial}{\partial t} + \mathbf{v} \cdot \nabla\right)\varepsilon = -p\nabla \cdot \mathbf{u} + \boldsymbol{\sigma} : \nabla\mathbf{v} - \frac{\varepsilon - \varepsilon_0}{\tau}, \tag{3}$$

$$\rho\left(\frac{\partial}{\partial t} + \mathbf{v} \cdot \nabla\right)\mathbf{B} = (\mathbf{B} \cdot \nabla)\mathbf{v} - (\nabla \cdot \mathbf{v})\mathbf{B}, \tag{4}$$

$$\nabla \cdot \mathbf{B} = 0. \tag{5}$$

The variables are the density $\rho$, the pressure $p$, the magnetic field $\mathbf{B}$, the plasma velocity field $\mathbf{v}$, the (uniform) gravitational acceleration $\mathbf{g}$, the ratio of specific heats $\gamma = 5/3$ and the specific energy density $\varepsilon = p/[(\gamma - 1)\rho]$. The associated temperature is given by $T = (\gamma - 1)\varepsilon$. The viscosity tensor $\boldsymbol{\sigma}$ is that of a Newtonian fluid[33]. Radiative effects in the atmosphere are modelled with the Newton cooling term in Eq. (3). Here, $\tau$ is the cooling time scale and $\varepsilon_0$ is the profile of the specific energy density at the start of the simulation.

The variables are made dimensionless with respect to photospheric values: pressure $p_{ph} = 1.4 \times 10^4$ Pa; density $\rho_{ph} = 2 \times 10^{-4}$ kg m$^{-3}$; speed $v_{ph} = 6.8$ km s$^{-1}$; scale height $H_{ph} = 170$ km; time $t_{ph} = 25$ s; surface gravity $g_{ph} = 2.7 \times 10^2$ m s$^{-2}$; magnetic field strength $B_{ph} = 1.3 \times 10^3$ G; temperature $T_{ph} = 5.6 \times 10^3$ K. The cooling time has a dimensionless value of $\tau = 0.5$.

The computational domain is split into four regions. Beneath the atmosphere is the convection zone ($z < 0$). Above the convection zone is a layer representing the photosphere/chromosphere ($0 \le z \le 10$). Above this lies the transition region ($10 < z \le 20$) and the corona ($z > 20$). Here, $z$ is the dimensionless height variable. The initial background atmosphere is in hydrostatic balance, with a plane-parallel temperature profile given by

$$T(z) = \begin{cases} 1 - \delta z \frac{\gamma - 1}{\gamma}, z < 0, \\ 1, 0 \le z \le 10, \\ 150^{\left[\frac{z-10}{10}\right]}, 10 < z \le 20, \\ 150, z > 20, \end{cases} \tag{6}$$

where $\delta$ is a nondimensional parameter that controls convection. For a convectively unstable layer that produces convection cells at the photosphere with properties close to those on the Sun, we take $\delta = 1.3$.

In order to excite convection, at $t = 0$, a perturbation in the $z$-component of the velocity is introduced and has the form

$$v_z = \frac{1}{10}e^{-(z+5)^2}\left[2\cos\left(\frac{2\pi x}{6\sqrt{3}}\right)\cos\left(\frac{2\pi y}{18}\right) + \cos\left(\frac{4\pi y}{18}\right)\right]. \tag{7}$$

This perturbation is also used in a previous study[33], though other similar perturbations can also produce the same required effect.

The initial magnetic field is given, in cylindrical coordinates, by

$$B_r = 0, B_y = B_0 e^{-\frac{r^2}{d^2}}, B_\theta = \alpha r B_y, \tag{8}$$

where $r^2 = x^2 + (z - z_{axis})^2$, $d$ is the flux tube radius, $B_0$ is the field strength at the axis of the tube and $\alpha$ is the uniform twist. For the simulation presented here, we take $z_{axis} = -15$, $d = 3$, $B_0 = 2.5$ ($3.25 \times 10^3$G) and $\alpha = 0.4$. To make the tube buoyant, a density deficit is introduced, with a profile of the form $\exp(-y^2/\lambda^2)$, where $\lambda = 20$.

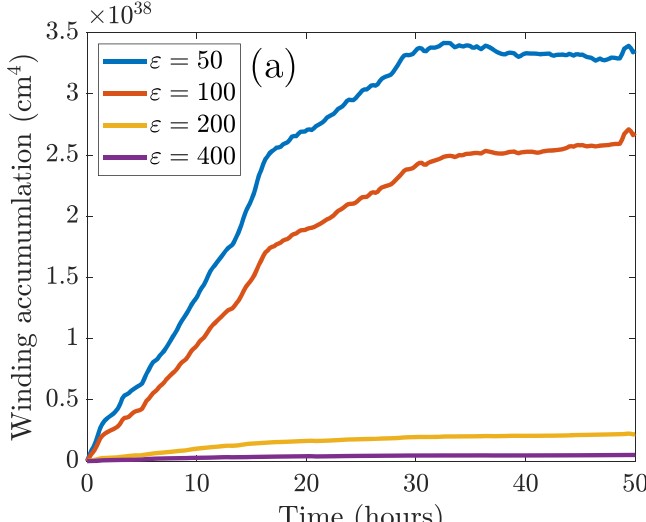

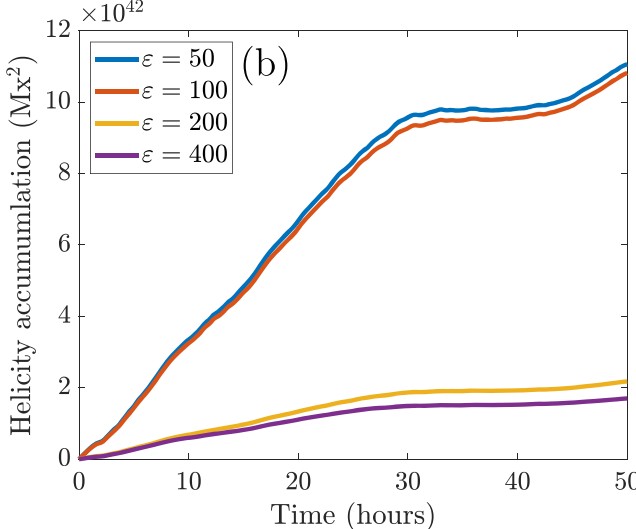

**Fig. 10 Cut-off sensitivity.** The accumulations for AR11318 starting from the end of the data gap (see Fig. 2). (**a**) shows the magnetic winding accumulations and (**b**) shows the magnetic helicity accumulations. Four different cut-offs $\varepsilon$ are considered (blue: $\varepsilon = 50$, red: $\varepsilon = 100$, yellow: $\varepsilon = 200$, purple: $\varepsilon = 400$). Source data are provided as a Source Data file.

The computational domain size is $(x, y, z) \in [-80, 80] \times [-80, 80] \times [-30, 80]$. There is a uniform resolution of $432^3$.

**Helicity and winding flux calculations**. The helicity flux through the photospheric boundary $P$ is given by

$$\frac{dH}{dt} = -\frac{1}{2\pi} \int_{P \times P} \frac{d\theta(\mathbf{x, y})}{dt} B_z(\mathbf{x}) B_z(\mathbf{y}) d^2 x \, d^2 y, \quad (9)$$

where $B_z$ is the component of the magnetic field $\mathbf{B}$ orthogonal to $P$, $\mathbf{x}$ and $\mathbf{y}$ are position vectors in $P$ that mark the intersection of field lines with the plane, and $\theta(\mathbf{x, y})$ is the angle made by the vector $\mathbf{x} - \mathbf{y}$ in $P$. The rate of change of this angle, measuring the rate of pairwise rotation of field lines, is given by

$$\frac{d\theta(\mathbf{x, y})}{dt} = \mathbf{e}_z \frac{(\mathbf{x} - \mathbf{y})}{|\mathbf{x} - \mathbf{y}|^2} \times \left( \frac{d\mathbf{x}}{dt} - \frac{d\mathbf{y}}{dt} \right), \quad (10)$$

where $\mathbf{e}_z$ is the unit vector orthogonal to $P$. In the formula above, the motion of a point $\mathbf{x}$, representing the intersection of a field line with $P$, is given by the field line velocity $\mathbf{u}$[18,19],

$$\mathbf{u(x)} = \frac{d\mathbf{x}}{dt} = \mathbf{v}_{||}(\mathbf{x}) - \frac{v_z(\mathbf{x})}{B_z(\mathbf{x})} \mathbf{B}_{||}(\mathbf{x}), \quad (11)$$

where $\mathbf{v}$ is the plasma velocity. The first term on the right-hand side is the in-plane velocity which braids the magnetic field in the atmosphere. The second term on the right-hand side is due to the emergence of the magnetic field, which causes apparent motion of the field line. The braiding contribution to helicity (and, similarly for the winding defined below) is found by setting $\mathbf{u} = \mathbf{v}_{||}$. Similarly, the emergence contribution is found by setting $\mathbf{u} = -v_z \mathbf{B}_{||}/B_z$.

The winding flux is given by

$$\frac{dL}{dt} = -\frac{1}{2\pi} \int_{P \times P} \frac{d\theta(\mathbf{x, y})}{dt} \sigma_z(\mathbf{x}) \sigma_z(\mathbf{y}) d^2 x \, d^2 y, \quad (12)$$

where the $\mathbf{e}_z$-component of the magnetic field is replaced by the indicator function

$$\sigma_z(\mathbf{x}) = \begin{cases} 1 & \text{if } B_z(\mathbf{x}) > 0 \text{ and } |B(\mathbf{x})| > \varepsilon, \\ -1 & \text{if } B_z(\mathbf{x}) < 0 \text{ and } |B(\mathbf{x})| > \varepsilon, \\ 0 & \text{if } B_z(\mathbf{x}) = 0 \text{ or } |B(\mathbf{x})| \le \varepsilon, \end{cases} \quad (13)$$

where $\varepsilon$ is a field strength cut-off[32,33]. The above integrals were determined numerically using a standard trapezoidal method. In this work, the winding inputs from the observations are calculated with a cut-off of $\varepsilon = 50$ G. Note that the 1-sigma error associated with SHARP data of the longitudinal magnetic field measurements is 10 G.

The sensitivity of the magnetic winding accumulation to the choice of cut-off is similar to that of the magnetic helicity accumulation, in the sense that both accumulations decay similarly as the cut-off is increased. An example of this is shown in Fig. 10, where both winding and helicity accumulations of AR11318 (starting from the end of the data gap) for different cut-offs are displayed.

As mentioned in the main text, plasma velocity $\mathbf{v}$ was obtained from observations using the DAVE4VM[35] method. The results presented were calculated with the version of the code written in Python but we have also checked the results using the version written in the Interactive Data Language (IDL). The SHARP vector magnetograms used for the presented results were downloaded using SunPy[39] from the Joint Science Operations Center (JSOC) database.

## Data availability

Source data are provided with this paper: observational helicity and winding accumulations are available from the data repository https://researchdata.gla.ac.uk/1197/, and are also available upon request from the corresponding author. The simulation data will not be stored due to their prohibitively large size, but are available from the corresponding author upon reasonable request. The instructions that accompany the simulation code (also found in https://researchdata.gla.ac.uk/1197/) will allow users to recreate the simulation and adapt it to consider other cases. The magnetogram data used in this study can be downloaded from http://jsoc.stanford.edu/ajax/lookdata.html.

## Code availability

The codes for reproducing the MHD simulation and the observational accumulation figures, together with instructions, are available from the data repository https://researchdata.gla.ac.uk/1197/. The codes will also be available from the corresponding author upon request. The Python version of the DAVE4VM can be found at https://github.com/Chicrala/pydave4vm.

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

## Acknowledgements

D.M., C.P. and B.R. welcome support from the US Air Force Office for Scientific Research (AFOSR): FA8655-20-1-7032. S.L.G. and P.R. welcome support from the Italian MIUR-PRIN grant 2017APKP7T "Circumterrestrial environment: Impact of Sun-Earth Interaction" and by the Istituto Nazionale di Astrofisica (INAF). This research received funding from the European Union's Horizon 2020 Research and Innovation programme under grant agreements No. 824135 (SOLARNET) and No. 729500 (PRE-EST).

## Author contributions

D.M. coordinated the project, co-developed the theoretical part of the work, performed the simulations, performed helicity and winding calculations, produced graphical output and drafted the manuscript; C.P. co-developed the theoretical part of the project, performed helicity and winding calculations and revised the manuscript; B.R. produced graphical output and revised the manuscript; P.R. performed helicity and winding calculations and revised the manuscript; S.L.G. selected the active regions for study and revised the manuscript. Numerical calculations were performed using the ARCHIE-WeSt High Performance Computer (www.archie-west.ac.uk) based at the University of Strathclyde (by D.M. and C.P.) and the DiRAC Extreme Scaling service at the University of Edinburgh (by D.M.), operated by the Edinburgh Parallel Computing Centre on behalf of the STFC DiRAC HPC Facility (www.dirac.ac.uk). This equipment was funded by BEIS capital funding via STFC capital grant ST/R00238X/1 and STFC DiRAC Operations grant ST/R001006/1. DiRAC is part of the National e-Infrastructure.

## Competing interests

The authors declare no competing interests.
