## [Peer Review File · Nature Communications]

REVIEWER COMMENTS

Reviewer #1 (Remarks to the Author):

GENERAL ASSESSMENT:

This study is based on an MHD simulation of an emerging twisted flux tube that creates a solar active region. The quality of the MHD simulation shown in Fig.1 is impressive. The authors address the fundamental question whether active regions can entirely be interpreted in terms of an emerging twisted flux tube. The authors demonstrate that the emergence accumulation dominates over the braiding accumulation, and thus conclude that it is due to a pre-twisted structure rather than due to photospheric footpoint motion.

While this issue is adequately addressed, the paper could be improved by toning down some over-reaching claims and by completing some ignored relevant literature. Otherwise I recommend the paper for publication in NCOMMS.

DETAILED COMMENTS:

There are a number of over-reaching claims, such as:

- line 14: "... there has not yet been any direct observational evidence of the emergence of large twisted magnetic flux tubes",

- line 15: "...the fundamental question, "are active regions formed by large twisted flux tubes?" has remained open."

- line 16: "...In this work, we answer this question in the affirmative and provide direct evidence to support this.

- line 19: "...This quantity, combined with other signatures that are currently available, provides the first direct evidence that large twisted flux tubes do emerge to create active region."

These statements ignore extensive previous work by other researches, who have demonstrated the emergence of twisted magnetic solenoids, flux tubes, and sigmoids. Work on the magnetic helicity using observational measurements have been accomplished by Chae (2001), Dalmasse et al. (2018), Demoulin et al. (2002), Green et al. (2002), Hagino and Sakurai (2004), Kusano et all (2002), LaBonte et al. (2007), Liu et al. (2014), Thalmann et al. (2011), Valori et al. (2016), Yang et al. (2013). The gaussian winding number has been measured recently by fitting nonlinear force-free magnetic field models (Aschwanden 2019). Some of the most relevant work deserves to be cited.

Fig.1a: Explain which MHD magnetic simulation code (MHD ?) has been used to generate this figure.

Fig.2: Define how the displayed winding accumulation relates to

the winding number. Can you provide a simple rule-of-thumb estimate of the winding accumulation value, given the fact that loops with winding numbers of >1 are kink unstable ? (Kliem, Toeroek etc.)

Reviewer #2 (Remarks to the Author):

This is a very interesting work, which represents a significant step towards understanding characteristics of magnetic flux emergence through the solar photosphere providing novel evidence that flux is emerging twisted. The method of separating magnetic braiding and winding fluxes has been developed and tested in simulations done by some of the authors of this paper, who now apply this for the first time to observational data of a small solar active region. Overall, I am happy with the paper, but I have a few concerns which I ask the authors to consider, discuss, or fix prior to publication.

Major concerns and questions:

- (i) The small (see my minor points below) AR analysed in this work lived for a mere few days, i.e. not much longer than the 80-hour period analysed in this work. Therefore, the period claimed to cover the emergence of this AR covers mostly its decay phase. This should be made clear in the paper.
- (ii) Strictly speaking, the emergence phase of this active region (AR) ended about 30 hours after its start, as at that time the AR reached its peak flux (Romano et al., 2014, your reference 23; their Figure 6, left panel). After that time, more than half of the winding flux shown in Figure 2 accumulated in the decay phase. This should be made clear in the paper.
- (iii) Now the question arises about the line drawn between the emergence and decay phases of an AR. The winding flux continues to rise for about 13-15 hours after peak magnetic flux has been reached. What does this mean for the flux emergence process, how does this fact fit in? How can the winding flux continue to rise while the total magnetic flux is beyond its peak and decreasing?
- (iv) The braiding flux starts a steeper rise after the peak magnetic flux has been reached and continues rising for 16-17 hours after the winding flux has reached its plateau. Would the rise of magnetic braiding flux be linked to the flux cancellation process along the magnetic inversion line of the AR, which has been in turn linked to the formation of the sigmoidal coronal structure, which takes place during the same period?

Minor points:

(i) In the 1st paragraph, you write: "This has led to observational proxies (such as sigmoidal field lines in the atmosphere⁴ and "magnetic tongue" patterns in magnetograms^{6,7})." The increase of electric currents in tandem with the increase of magnetic flux is another observational proxy which could be mentioned. This was first shown by Leka et al. (1996, ApJ 462, 547). On the other hand, sigmoidal coronal structure typically forms during the decay phase of active regions, so it is misleading to mention it together with the photospheric magnetic tongue structure, which is a flux-emergence phase twist proxy.

(ii) At the end of the 1st paragraph, you mention shearing motions to form sigmoids. Again, sigmoids mainly form during the decay phase of active regions, not in the emergence phase. They may form, however, in between two adjacent emerging-flux regions when opposite-polarity spots belonging to different emerging/diverging dipoles reconnect and shearing motions are provided by the divergence of each dipole's opposite polarities. This was the case in AR 11158 (February 2011), but in that case the shearing motion was not taking place within the emerging dipole itself.

(iii) In the 3rd paragraph, which starts with "Magnetic winding", you mention serpentine fields. Serpentine fields are usually thought to be emerging field lines crossing the photosphere by the assistance of convection and reconnection, not field lines which have emerged and then dragged down by convection. Are you sure that the latter is seen in simulations?

(iv) Line 114: Although larger than the simulated AR, having a peak magnetic flux of 1.5×10^{21} Mx (as shown by Romano et al., 2014, your reference 23), AR 11318 is considered a small AR. (Cf. the point you made in the former paragraph).

(v) Figure 3: Are the winding and braiding flux distribution patterns, shown here at around (somewhat after) the peak flux has been reached in the AR, representative of the later times, e.g. when the winding-flux plateau is reached? Please briefly describe how the pattern is changing.

Reviewer #3 (Remarks to the Author):

Review of the paper entitled "Direct evidence: twisted flux tube emergence creates solar active regions" by MacTaggart et al.

The paper presents a new method to quantify the emergence of twist through the solar photosphere. To this end the paper introduces a topological quantity called "magnetic winding", which unlike helicity is not quadratic in field strength. The paper claims that this analysis provides the first direct evidence for the formation of active regions from emerging twisted flux tubes.

The paper introduces a new topological quantity, the "winding flux", which is essentially the magnetic helicity flux normalized such that it does no longer depend on the magnetic field strength. The paper claims that this is a more useful quantity than the helicity flux as it clearly separates field line topology from magnetic flux, although this claim is not well supported by analysis or additional evidence. While this methodology is applied to a numerical simulation of emerging twisted flux, that alone does not prove that it is a better approach to characterize emergence of twisted flux -- Fig 1 and the associated analysis merely show that these quantities can be computed from a (rather simple) simulation. Supporting this claim would require here a more thorough study of flux emergence simulations including also more complex topologies that would produce more likely ambiguous results in the analysis.

On a fundamental level this comes down to the question of what a "twisted flux tube" is as a physical object. For good reasons the name contains "twist" and "flux" together, so relying on a method that only stresses the "twist" is as incomplete as one only capturing "flux". There is no expectation that in a complex flux tube twist is constant over the cross-section of the tube. Helicity flux provides more naturally a field strength weighting centered towards the core of the flux tube, while the proposed winding flux weights all photospheric pixels equally regardless of their association with emerging flux. A complete picture requires a combination of all these measures. The fact that the helicity flux may produce in certain cases other results than the winding flux does potentially indicate rather complicated flux structures, but it does not mean that the winding flux is necessarily the superior measure – at least the paper did not demonstrate that.

Another concern is the influence of measurement errors on the determination of these quantities. The helicity flux does have a natural bias towards stronger fields, with more accurate measurements (including disambiguation, projection effects and rotation from the longitudinal-transverse to x,y,z directions), while the winding flux is based on data with generally weaker field and more complex small-scale structures. While the authors use the cutoff at a 50G level to counter some of these effects, a more careful error analysis may be warranted here. There is no mention of the position of the analyzed active regions on the solar disk and how that may affect the determination of the derived quantities. Fig 3 appears to indicate that the winding flux is noisy and shows some rather sharp patch-like features that do not look very natural.

Minor points:

The color tables in Fig 3 and extended data Fig 2 could be improved. I would suggest a color scheme that diverges at the zero level (i.e. red-blue with different colors for extreme values).

Is there a way to put the helicity and winding accumulations on a comparable scale, i.e. through a normalization with an integral of B_z^2 over the photospheric boundary?

To summarize, the paper does introduce an interesting new quantity that may help quantifying the emergence of twisted flux better than helicity alone. However, the presentation in the current paper does not provide the evidence to justify strong claims such as " In this work, we answer this question in the affirmative and provide direct evidence to support this.", "This quantity, combined with other signatures that are currently available, provides the first direct evidence that large twisted flux tubes do merge to create active regions." However, I do agree that the magnetic winding does provide additional useful evidence to complete the picture, and I strongly encourage the authors to refine their methodology.

Referee 1

GENERAL ASSESSMENT:

This study is based on an MHD simulation of an emerging twisted flux tube that creates a solar active region. The quality of the MHD simulation shown in Fig.1 is impressive. The authors address the fundamental question whether active regions can entirely be interpreted in terms of an emerging twisted flux tube. The authors demonstrate that the emergence accumulation dominates over the braiding accumulation, and thus conclude that it is due to a pre-twisted structure rather than due to photospheric footpoint motion. While this issue is adequately addressed, the paper could be improved by toning down some over-reaching claims and by completing some ignored relevant literature. Otherwise I recommend the paper for publication in NCOMMS.

We thank the referee for a positive assessment of our work. Although we use a simulation as an example (as we have done in previous works: Prior & MacTaggart 2019, *J. Plasma Phys.* **85**, 775850201; MacTaggart & Prior 2020, *GAFD*, **115**, 85), the study is based on a fundamental geometrical result, which we have treated theoretically (see MacTaggart & Prior 2020, *J. Phys. Conf. Ser.*, 1730; Prior & MacTaggart 2020, *Proc. R. Soc. A*, **476**, 20200483), in simulations and now, in this work, in observations. The simulations allow us to test that our result is sound when solving the full MHD equations in a stratified atmosphere with convection.

With respect to “toning down some over-reaching claims”, we are happy to make changes to the language whilst still highlighting the importance of the result. With respect to “ignored relevant literature”, before transferring to *Nature Communications*, we had an imposed limit of 30 references in the main text. This restriction has now been lifted and we are more than happy to include more citations to relevant literature.

DETAILED COMMENTS:

There are a number of over-reaching claims, such as:

- line 14: "... there has not yet been any direct observational evidence of the emergence of large twisted magnetic flux tubes",

- line 15: "...the fundamental question, "are active regions formed by large twisted flux tubes?" has remained open."

- line 16: "...In this work, we answer this question in the affirmative and provide direct evidence to support this.

- line 19: "...This quantity, combined with other signatures that are currently available, provides the first direct evidence that large twisted flux tubes do emerge to create active region."

We are happy to follow the referee’s advice and edit the language of the above statements. The above text has been updated in the new manuscript (see lines 9 – 20). Although we edit the above text, toning its directness down, we maintain that our methodology and results are new and represent an advance, building on previous work.

These statements ignore extensive previous work by other researches, who have demonstrated

the emergence of twisted magnetic solenoids, flux tubes, and sigmoids. Work on the magnetic helicity using observational measurements have been accomplished by Chae (2001), Dalmasse et al. (2018), Demoulin et al. (2002), Green et al. (2002), Hagino and Sakurai (2004), Kusano et al. (2002), LaBonte et al. (2007), Liu et al. (2014), Thalmann et al. (2011), Valori et al. (2016), Yang et al. (2013). The gaussian winding number has been measured recently by fitting nonlinear force-free magnetic field models (Aschwanden 2019). Some of the most relevant work deserves to be cited.

As mentioned above, we are now in a position to expand our citation list. We would be hesitant to endorse the strong statement that these works “have demonstrated the emergence of twisted magnetic solenoids, flux tubes and sigmoids.” Some of these papers (as well as others not listed) form important precursors to our work and are, therefore, rightly worthy of citation. However, we would like to make clear that, although providing important advances, they do not provide the direct evidence of twisted flux tube emergence that the magnetic winding provides. We now go through each of the listed papers to discuss briefly their results in connection with our work.

Chae 2001: We have cited this work in our previous publications on helicity and winding input. The interpretation of the changing sign of the helicity injection signature was intriguing to us and one of the initial prompts for us to investigate this subject.

Dalmasse et al. 2018: This is a very interesting paper about a method of relating connectivity to helicity. There is a fundamental difference between their work and ours, however. In creating connectivity metrics, they normalize relevant quantities outside an integral (e.g. see their equations 11,12), whereas we perform the renormalization inside the integral (for the magnetic winding). The difference is that magnetic winding suffers no weighting due to the magnetic field which, as mentioned our the paper (see also Prior & MacTaggart 2020, Proc. R. Soc. A, **476**, 20200483), has a large influence on the measured quantity. Thus, we would not expect their method to produce as clear a signal for twisted tube emergence due to the influence of the magnetic field weighting in their metrics (though we would be happy to be proved wrong and have another method to compare with directly).

Demoulin et al. 2002: This important work assesses the contribution of horizontal shear flows, including differential rotation, to helicity injection. They conclude that helicity injection due to shear flows is small compared to the helicity of a twisted flux tube (which could emerge to form an active region). Although this picture develops in later works on helicity (see the ratio between the braiding and emergence contributions in our work and others), their work shows that the emergence of pre-twisted magnetic field can be an important factor for the helicity injection. Our work provides the evidence that pre-twisted tubes do emerge and so our work complements the work of Demoulin et al. well.

Green et al. 2002: They measure the helicity input due to differential rotation and conclude that it is not enough to explain the helicity released by CMEs. Although we do not focus on CMEs in our work, the evidence that we provide, namely that pre-twisted tubes emerge, shows that there is an additional source of helicity to shearing motions, which, combined with the contribution from differential rotation, may fill the helicity deficit that they report. This, however, is a question for a different study.

Hagino and Sakurai 2004: They study the α (a measure of the local twist of magnetic field) as a source of helicity. Since this quantity is local, however, it is not as robust as magnetic

winding (a global measure). Consider the following example. When the α of a potential arcade is measured on a plane, the value is zero. If there is a small perturbation to that plane, the α signal changes, giving a false signal. This is illustrated in Figure 1.

Figure 1: The left panel shows a potential arcade and a photospheric boundary with a perturbation. The right panel shows the α distribution calculated without treating the perturbation correctly. The value should be zero everywhere for a potential field but is non-zero due to the perturbation.

Magnetic winding, being a global quantity, is more robust to such changes (see Section 3(c) of Prior & MacTaggart 2020, Proc. R. Soc. A, **476**, 20200483). Therefore, the interpretation of α values must be treated very carefully. For the application in their paper, we think that the approach is fine. However, it is significantly different from our measure of magnetic winding.

Kusano 2002: In this work, both the braiding and emergence contributions to helicity injection are considered. Further, both are found to be of importance. This ties in well with our study. What we do that is different, however, is give specific detail on the magnetic topology of the emerging field. For the “clean” bipolar cases considered, we can find a clear signature for a twisted flux tube in the winding injection, something that is not possible, in general, for the helicity injection.

LaBonte et al. 2007: This is an extensive survey on the relationship between helicity injection and X-class flares. In their survey, they determine a lower bound on the helicity required for strong flares. This study is not, however, concerned with the magnetic topology of emerging fields, as we are. Although the helicity input is important, without the magnetic winding, it is not clear which part of the helicity dominates (the field line entanglement or the field strength). Further, they assume that the field line velocity and the local correlation tracking velocity are equivalent. Doing this, they focus on the braiding contribution (which may be relevant for X-class flares, based on the standard flare model).

Liu et al. 2014: They find that the braiding contribution dominates the emergence contribution for helicity injection, just as we do. The magnetic winding gives extra information, allowing us to identify if a twisted flux tube emerges (the importance of the emergence and braiding contributions switches).

Thalmann et al. 2011, Valori et al. 2016, Yang et al. 2013: These works are related to the

calculation of relative helicity for 3D fields. Since our focus is on calculating fluxes, which can be found directly from observations, these works are not strictly relevant to ours.

Aschwanden 2019: This is an interesting paper which considers a local form of winding. Although somewhat different from our global measure of generalized winding (i.e. the average of all pairs of field lines in the domain winding about each other) which is evolving in time (we measure the flux of a quantity, so winding could increase/decrease with emergence/submergence), this article is worth citing as an example of the application of winding in practice.

After assessing the above works from the literature, we have cited the following in our paper: Chae 2001, Demoulin et al. 2002, Green et al. 2002, LaBonte et al. 2007, Liu et al. 2014 and Aschwanden 2019. The inclusion of extra references, with updated text, can be found on lines 32 – 36 and 43.

Fig.1a: Explain which MHD magnetic simulation code (MHD ?) has been used to generate this figure.

We use the open-source Lare3d code to solve the compressible MHD equations (this is mentioned in the Methods section). The figure was created using the open-source visualization tool Visit. We will include a version of the code with details on how to set up the simulation used in the paper (and variations on this). Details of the simulation can be found in MacTaggart & Prior 2020, GAFD, **115**, 85.

Fig.2: Define how the displayed winding accumulation relates to the winding number. Can you provide a simple rule-of-thumb estimate of the winding accumulation value, given the fact that loops with winding numbers of > 1 are kink unstable? (Kliem, Toeroek etc.)

We have searched to find a simple rule-of-thumb for the winding accumulation but, unfortunately, we have not yet found one. In MacTaggart & Prior 2020, GAFD, **115**, 85, we investigated both helicity and winding in emerging twisted flux tubes, changing only the initial field strength. Since winding is a topological quantity integrated over an area, the different expansions of the tubes during the emergence process seem to prevent any simple rule-of-thumb, even if they all produce an accumulation signature that is qualitatively the same. We will continue to search for simplified rules-of-thumb.

We thank Referee 1 again for valuable comments that have helped to improve the paper. In particular, our method and results now sit more soundly in the context of previous work, without watering down the advance presented in our work. We hope that our changes are to the referee's satisfaction.

Yours sincerely,

D. MacTaggart, C. Prior, B. Raphaldini, P. Romano and S.L. Guglielmino

Referee 2

This is a very interesting work, which represents a significant step towards understanding characteristics of magnetic flux emergence through the solar photosphere providing novel evidence that flux is emerging twisted. The method of separating magnetic braiding and winding fluxes has been developed and tested in simulations done by some of the authors of this paper, who now apply this for the first time to observational data of a small solar active region. Overall, I am happy with the paper, but I have a few concerns which I ask the authors to consider, discuss, or fix prior to publication.

We thank the referee for a positive assessment of our work.

Major concerns and questions:

(i) The small (see my minor points below) AR analysed in this work lived for a mere few days, i.e. not much longer than the 80-hour period analysed in this work. Therefore, the period claimed to cover the emergence of this AR covers mostly its decay phase. This should be made clear in the paper.

We have updated the description of what the 80-hour period covers (lines 141 – 142). We have not referred explicitly here to a “decay phase” but refer to this later in lines 227 – 246. However, please see our response to (iii) first.

(ii) Strictly speaking, the emergence phase of this active region (AR) ended about 30 hours after its start, as at that time the AR reached its peak flux (Romano et al., 2014, your reference 23; their Figure 6, left panel). After that time, more than half of the winding flux shown in Figure 2 accumulated in the decay phase. This should be made clear in the paper.

This has also been made clear in the paper (lines 230 – 231). Please see our response to (iii) below for more discussion related to this.

(iii) Now the question arises about the line drawn between the emergence and decay phases of an AR. The winding flux continues to rise for about 13-15 hours after peak magnetic flux has been reached. What does this mean for the flux emergence process, how does this fact fit in? How can the winding flux continue to rise while the total magnetic flux is beyond its peak and decreasing?

For this AR, there are two important considerations to take into account. The first is specific to this particular AR, which we chose as a “clean” example that could be compared (within reason) to simulations that we have performed. Unfortunately, there are some missing data for this AR. Although this does not affect our main result, it can have a quantitative effect on how the winding accumulation appears. If, as argued in the paper (lines 158 – 161), the accumulation of all the winding components behaves in a linear fashion during the data gap, then the emergence contribution would be even larger compared to the braiding contribution

than it is now. This would have an effect on the total winding, which would look more closely like the emergence contribution.

If, however, the data were as they are without any missing segments, the issue is not controversial. This is because there need not be a one-to-one time correspondence between magnetic winding and magnetic flux due to the nature of flux emergence. The following description can be corroborated by many different studies of flux tube emergence (see, for example, the reviews Hood et al. 2012, *Solar Phys.*, **278**, 3; Cheung & Isobe 2014, *Living Rev. Sol. Phys.* **11**, 3). When a twisted flux tube reaches the photosphere, it cannot continue to rise due to buoyancy and essentially becomes stuck there. The sunspots can develop but the bulk of the (near-horizontal) twisted tube between the spots remains trapped at or just below the photosphere. This part of the magnetic field can later expand into the corona via a buoyancy instability. This leads to the transference of twist into the atmosphere. An example of this from a MHD simulation of flux emergence is shown in Figure 1.

Figure 1: Taken from Hood, Archontis and MacTaggart 2012, *Solar Phys.*, **278**, 3. The contour gives the magnitude of the magnetic field at normalised times 20 (top left), 40 (top right), 60 (bottom left) and 80 (bottom right). The grey region indicates the solar interior out to one pressure scale height below the base of the photosphere.

Thus, there can be a temporal gap between the traditional “emergence phase” described by the (sunspot-dominated) magnetic flux and the emergence of horizontal parts of the magnetic field (which dominate the winding accumulation). It is for this reason that we did not make a strong identification, based on the magnetic flux, of an “emergence phase” and a “decay phase.” We

have included a discussion of this point in lines 227 – 246.

(iv) The braiding flux starts a steeper rise after the peak magnetic flux has been reached and continues rising for 16-17 hours after the winding flux has reached its plateau. Would the rise of magnetic braiding flux be linked to the flux cancellation process along the magnetic inversion line of the AR, which has been in turn linked to the formation of the sigmoidal coronal structure, which takes place during the same period?

Comparing with Figure 1 (I,J) of Romano et al. 2014, the rise in the braiding contribution does appear to coincide with the development of the sigmoid. We have not focussed primarily on this aspect on this paper and have plans to consider it in more detail in future work. However, we can confirm that the braiding contribution is due to vortical flow near the sunspots which leads to the shearing of the magnetic field.

Minor points:

(i) In the 1st paragraph, you write: “This has led to observational proxies (such as sigmoidal field lines in the atmosphere⁴ and “magnetic tongue” patterns in magnetograms^{6,7}).” The increase of electric currents in tandem with the increase of magnetic flux is another observational proxy which could be mentioned. This was first shown by Leka et al. (1996, ApJ 462, 547). On the other hand, sigmoidal coronal structure typically forms during the decay phase of active regions, so it is misleading to mention it together with the photospheric magnetic tongue structure, which is a flux-emergence phase twist proxy.

We agree that Leka et al. 1996 is an important paper to cite in our work and we have included this as an example of a paper which concludes that pre-twisted magnetic fields should emerge (see line 33). Our work complements the results of that paper very well. Please note that before transferring to *Nature Communications*, we had a limit of 30 references in the main text. This has now been lifted so we have expanded our citations of the literature.

A sigmoid, being a manifestation of current (and, therefore, twist) in a bipolar magnetic field, is a natural consequence of the injection of twist into the atmosphere by an emerging twisted tube. Therefore, although the sigmoid itself may form in the decay phase of the AR, its origin is due to the emergence phase. Thus, we feel it is reasonable to include this a proxy of twisted tube emergence.

(ii) At the end of the 1st paragraph, you mention shearing motions to form sigmoids. Again, sigmoids mainly form during the decay phase of active regions, not in the emergence phase. They may form, however, in between two adjacent emerging-flux regions when opposite-polarity spots belonging to different emerging/diverging dipoles reconnect and shearing motions are provided by the divergence of each dipole’s opposite polarities. This was the case in AR 11158 (February 2011), but in that case the shearing motion was not taking place within the emerging dipole itself.

It is true that sigmoids can occur between active regions. Our point, however, was that if a twisted tube emerges to create an AR, a sigmoid will be a natural consequence (see, for example, Archontis et al. 2009, ApJ, **691**, 1276).

(iii) In the 3rd paragraph, which starts with “Magnetic winding”, you mention serpentine fields. Serpentine fields are usually thought to be emerging field lines crossing the photosphere by the assistance of convection and reconnection, not field lines which have emerged and then dragged down by convection. Are you sure that the latter is seen in simulations?

Yes, the latter is seen in simulations. We have performed flux emergence simulations with convection that show the dragging down of field lines (this has important consequences for the helicity flux, see MacTaggart & Prior 2020, *GAFD*, **115**, 85). Other studies that show this clearly are Tortosa-Andreu & Moreno-Insertis 2009, *A&A*, **507**, 949 and Cheung et al. 2010, *ApJ*, **720**, 233.

(iv) Line 114: Although larger than the simulated AR, having a peak magnetic flux of 1.5×10^{21} Mx (as shown by Romano et al., 2014, your reference 23), AR 11318 is considered a small AR. (Cf. the point you made in the former paragraph).

We have made it clear in the paper that this is a small region (see line 135) - though this point does not affect our main result.

(v) Figure 3: Are the winding and braiding flux distribution patterns, shown here at around (somewhat after) the peak flux has been reached in the AR, representative of the later times, e.g. when the winding-flux plateau is reached? Please briefly describe how the pattern is changing.

At later times the magnetic flux begins to disperse and consequently the winding input pattern (particularly that of the emergence component) also disperses (the flux tube has emerged and there is no other emergence that transports topologically significant field into the atmosphere). As an example, Figure 2 shows the distributions of the two components of the winding inputs at $t = 78$ hours, a time far into the plateau phase. We have now included this figure in the updated manuscript (Figure 4 in the manuscript, lines 180 – 183).

We thank Referee 2 again for valuable comments that have helped to improve the paper. We hope that our changes are to the referee’s satisfaction.

Yours sincerely,

D. MacTaggart, C. Prior, B. Raphaldini, P. Romano and S.L. Guglielmino

Figure 2: The braid and emergence winding distributions at $t = 78$ hours.

Referee 3

We thank the referee for taking the time to review our paper. Below are the responses to all questions and concerns.

The paper introduces a new topological quantity, the “winding flux”, which is essentially the magnetic helicity flux normalized such that it does no longer depend on the magnetic field strength. The paper claims that this is a more useful quantity than the helicity flux as it clearly separates field line topology from magnetic flux, although this claim is not well supported by analysis or additional evidence. While this methodology is applied to a numerical simulation of emerging twisted flux, that alone does not proof that it is a better approach to characterize emergence of twisted flux – Fig 1 and the associated analysis merely show that these quantities can be computed from a (rather simple) simulation. Supporting this claim would require here a more thorough study of flux emergence simulations including also more complex topologies that would produce more likely ambiguous results in the analysis.

In our paper, we use magnetic winding to provide clear details about the emerging magnetic topology, which the magnetic helicity cannot provide. However, we would like to stress that we do not consider winding to be “more useful” than helicity in a general sense. Both quantities give different information and they are most useful when calculated together (see lines 73 – 78). We also stress that, although we use a simulation as an example (as we have done in previous works: Prior & MacTaggart 2019, *J. Plasma Phys.* **85**, 775850201; MacTaggart & Prior 2020, *GAFD*, **115**, 85), the study is based on a fundamental geometrical result, which we have treated theoretically (see MacTaggart & Prior 2020, *J. Phys. Conf. Ser.*, 1730; Prior & MacTaggart 2020, *Proc. R. Soc. A*, **476**, 20200483), in simulations and now, in this work, in observations. The simulations allow us to test that our result is sound when solving the full MHD equations in a stratified atmosphere with convection.

In reference to “including more complex topologies,” this is something that we have considered in previous publications cited within the paper. Before describing these, we briefly describe the simulation we presented as an example. Our domain stretches from the convection zone to the corona. Convection is driven so that the plasma velocities and cell sizes at the photosphere are close to those on the Sun. A twisted tube is placed in the convection zone and rises buoyantly to the photosphere. During this rise it is deformed significantly and upon reaching the photosphere, the field expands into the higher atmosphere. The simulation contains all the necessary elements needed for a thorough analysis of the injection of magnetic winding and helicity. Extensive details of the emergence of twisted flux tubes of different field strengths can be found in MacTaggart & Prior 2021, *GAFD*, **115**, 85. This work shows that the winding injection signature for twisted flux tubes, across all the simulations ran, is qualitatively the same, whereas that of helicity injection changes between the cases. Let us stress again that this does not make winding more useful, the different signatures give different and important information.

In Prior & MacTaggart 2019, *J. Plasma Phys.* **85**, 775850201, we considered both twisted and “mixed helicity” tubes. A mixed helicity tube is one which is globally untwisted but has an equal and opposite number of positive and negative twists. Thus, when it emerges, it inputs zero helicity overall into the atmosphere. In our 2019 paper, our computational model was simpler in

the sense that the background atmosphere was convectively stable. The winding inputs varied between the twisted and mixed helicity tubes. This was because of two things. In the mixed helicity tubes, the small twist regions can emerge at different times, causing a more undulating winding accumulation. Further, as the emerging part of the field is very weakly twisted overall, this causes buckling of the field lines leading to submergence. This has an important effect on the winding accumulation, where there can be an $O(1)$ change reversing the sign of the input. In an earlier paper (Prior and MacTaggart 2016, GAFD, **110**, 432), although we did not calculate winding explicitly, this behaviour can also be seen in the evolution of emerging mixed helicity fields.

In running flux emergence simulations with convection for mixed helicity fields (at the same length scale as the simulation we present in the paper), we find that, unlike in the convectively-stable case, the tubes are deformed so much that they may not reach the photosphere. Those that do emerge, do so in small patches and no longer represent a viable model for active regions. This result is not really a surprise since without an overall twist, tubes in a convectively-unstable layer are known to quickly break up (see cited works in our paper). It may be the case that mixed helicity tubes that are much larger than those we have modelled may survive to emerge. If so, however, based on our previous work described above, we would expect very different winding accumulations compared to the twisted tube case.

In response to the referee’s comment, we have also simulated cases of twisted tubes with mixed helicity components, i.e. containing an equal and opposite number of localized positive and negative twists. The magnetic field’s initial total helicity and winding values are similar to that presented in our manuscript, but that they have a more complex small scale distribution. For those simulations that lead to a simple bipolar emergence, the signature does not change from that described in the paper, and we do not show these for brevity. Instead here, we focus on a case where the tube emerges in multiple locations, thus potentially creating more complexity in the winding signature. Figure 1 (a) displays the winding accumulation for an emerging tube that initially has a constant twist with two cancelling mixed helicity perturbations. (b) shows an isosurface of the tube at a specific time.

The effect of multiple regions of the partial emergence of the tube is to cause very small plateaus in the winding accumulation signature. However, the global signature still follows that in the paper. The isosurface is just before the first of the “partial emergences” and indicates the serpentine nature of the emergence. Since all of the sub-parts of the magnetic field emerge within a short time of each other, there is, overall, the “rise and plateau” signature. Each partial emergence adds its own “rise and plateau” signature.

Figure 2 shows a magnetogram (B_z) at $t = 1800$ s, during the winding plateau. This “serpentine” case produces no simple bipolar structure, yet the magnetic winding still detects the correct topological input.

One important question to ask is, why is there no obvious effect on the winding due to mixed helicity perturbations? The answer can be found by considering what happens to the flux in the convection zone and what the magnetic winding measures. In the convection zone, the usual image of a twisted flux tube disappears due to the deforming effects of convection (as seen in Figure 1(a) of our paper). That being said, the topology of the magnetic field remains, on the whole, intact. That is, despite the tube being deformed beyond the visual recognition of a twisted tube, it is still an *oriented* and *compact* magnetic field with a *net value of helicity*. The

small mixed helicity perturbations add no net helicity to the tube, so their effect compared to the deforming effects of convection (which can twist field lines locally) is marginal. In particular, they have almost no effect on the magnetic winding accumulation, which is a *global* measure of the magnetic topology (see lines 95 – 103; 121 – 125).

In summary, we have performed, over the past few years, purely geometrical calculations of an emerging twisted field (Prior & MacTaggart 2020, Proc. R. Soc. A.), twisted flux emergence simulations (MacTaggart & Prior 2021, GAFD and here) and the mixed helicity/twisted simulations described above. These all had an initial net helicity/winding and a twisted core, even if the whole structure is distorted/has additional small scale complexity. All examples yield a rise then plateau signature indicative of the emergence of this pre-existing net helicity/winding through a surface. The presence of this signature in observational winding is a clear indication of the emergence, in a real active region, of pre-existing entanglement with a twisted core. The fact the helicity signatures in these calculations (both simulations and observations) vary significantly (see MacTaggart & Prior 2021, GAFD) is an indication that the winding calculations of such systems provides significant *additional* and *clarifying* information which can complement the existing array of magnetic field diagnostics of active region evolution. This is our claim in the paper (see the Discussion) and we are confident it is fully justified. We have included the serpentine simulation, described here, as Supplementary Information. This adds further evidence of the robustness of the winding measure. We will release our simulation code with detailed instructions, allowing others to generate cases of their own for testing.

On a fundamental level this comes down to the question of what a "twisted flux tube" is as a physical object. For good reasons the name contains "twist" and "flux" together, so relying on a method that only stresses the "twist" is as incomplete as one only capturing "flux". There is no expectation that in a complex flux tube twist is constant over the cross-section of the tube. Helicity flux provides more naturally a field strength weighting centered towards the core of the flux tube, while the proposed winding flux weights all photospheric pixels equally regardless of their association with emerging flux. A complete picture requires a combination of all these measures. The fact that the helicity flux may produce in certain cases other results than the winding flux does potentially indicate rather complicated flux structures, but it does not mean that the winding flux is necessarily the superior measure – at least the paper did not demonstrate that.

Our previous “definition” of a realistic twisted flux tube, namely a compact and directed magnetic field with a net value of helicity, holds here (see lines 99 – 101). The simple cartoon of a flux tube with uniform twist has to go. As mentioned above, however, deformation of the tube does not disrupt its global magnetic topology (at least not significantly). This global net helicity allows the tube to remain, overall, intact as it rises through the convection zone. When the tube emerges into the atmosphere, the magnetic field, despite being predominantly bipolar, is complex. There are patches with different field strength and different local field line windings. The helicity flux cannot always identify the global magnetic topology as its signal is sensitive to local variations in the field strength (and particularly the difference between the sunspots and the rest of the active region, see the discussion in our paper). This result is clear from our previous study of different field strengths (MacTaggart & Prior 2021, GAFD, **115**, 85). By removing the flux weighting, the magnetic winding can detect the global topology.

The magnetic winding weights all photospheric pixels equally *subject* to a tolerance measure applied to the magnetic field (more on this later for the next point). We have deliberately chosen “clean” (isolated and bipolar) active regions for this study in order to present as clear a signal as possible for the winding flux. It is also important to highlight, as indicated in our answer to the referee’s next comment, that magnetic flux and magnetic helicity are both dominated by field values (pixels) of magnitudes between 100-200 G. The difference being that for the helicity input this value is dominated by the field’s B_z component whilst for the winding it is those with significant in-plane components. Both are critical components of the field but inform on differing parts of the active region. Both are important sources of information and could complement each other when used in tandem.

Thus we would like to reiterate that we do not consider magnetic winding to be superior to magnetic helicity. The latter is a fundamental invariant of MHD. The former is related to helicity but can give different information (see Prior & MacTaggart 2020, Proc. R. Soc. A, **476**, 20200483 for a detailed discussion of this). We have tried to clarify in the paper that magnetic winding is an important addition to magnetic helicity which can provide new information, rather than being a replacement for it (lines 73 – 78).

Another concern is the influence of measurement errors on the determination of these quantities. The helicity flux does have a natural bias towards stronger fields, with more accurate measurements (including disambiguation, projection effects and rotation from the longitudinal-transverse to x,y,z directions), while the winding flux is based on data with generally weaker field and more complex small-scale structures. While the authors use the cutoff at a 50G level to counter some of these effects, a more careful error analysis may be warranted here. There is no mention of the position of the analyzed active regions on the solar disk and how that may affect the determination of the derived quantities. Fig 3 appears to indicate that the winding flux is noisy and shows some rather sharp patch-like features that do not look very natural.

We have re-performed the helicity and winding flux calculations with a variety of different cut-offs. These are shown in Figure 3 (of these responses) where, for ease of comparison, we show the integrated accumulations after the time period of missing data for AR11318.

It is clear that up to 100G, there is a small quantitative change in both the helicity and winding accumulations (with a slightly larger relative difference for the winding) and no qualitative change. Between 100-200G, both accumulations drop by an order of magnitude. Although 200G is a very large cut-off, this result shows that the helicity and winding accumulations are affected by cut-offs in a similar way. Thus, both measures appear to have similar sensitivities to what field strengths are included in the calculations (which is what we should expect as both are related measures of the same magnetic field). This is highlighted in Figures 4 (for the winding) and 5 (for the helicity) which display the regions of strong transverse field and strong vertical field at $t = 45$ hours respectively (in panels (a)). As expected, the strong vertical field matches well with the sunspots and the strong horizontal field is prominent at the penumbra and between the spots. In both cases there is a clear correlation between the field distributions and their concurrent winding/helicity densities (panels (b)). This serves to further highlight the fact the winding gives preference to topological information in a different part of the emerging field from the helicity distribution, and hence adds complementary but new information not available from the helicity calculation alone.

The used cut-offs appear to ensure that noisy distributions due to small-scale weak emerging fields are not affecting the measure of the global topology of the emerging active region. Further, we selected active regions emerging between $\pm 30^\circ$ from the central meridian, in order to minimize the projection effects and, as much as possible, the uncertainties induced by the disambiguation algorithm used in the SHARP data. In particular, AR11318 emerged at -15° from the central meridian. Edits to the paper, related to the above issues, can be found on lines 140 and 318 – 325).

Minor points:

The color tables in Fig 3 and extended data Fig 2 could be improved. I would suggest a color scheme that diverges at the zero level (i.e. red-blue with different colors for extreme values).

This change has been made.

Is there a way to put the helicity and winding accumulations on a comparable scale, i.e. through a normalization with an integral of B_z^2 over the photospheric boundary?

For AR11318, we have multiplied the winding calculations by the spatially averaged value of B_z^2 ($\langle B_z^2 \rangle$) for each time (its value is shown in Figure 6(a)). It does not have a significant effect on the basic shape of the winding input save to alter the (relative) steepness of the early input phase. In Figure 6(b) we plot the ratio $H/(L\langle B_z^2 \rangle)$. Unsurprisingly it increases during the phase in which the emergence contribution to the winding has plateaued. During this period the helicity, dominated by the braiding contribution continues to rise. Whilst it is intriguing to see that $L\langle B_z^2 \rangle$ remains the same order of magnitude as the helicity, we don't think it adds any extra information not already in the plots. For this reason, and to save space, we do not include it in updated paper.

To summarize, the paper does introduce an interesting new quantity that may help quantifying the emergence of twisted flux better than helicity alone. However, the presentation in the current paper does not provide the evidence to justify strong claims such as " In this work, we answer this question in the affirmative and provide direct evidence to support this.", "This quantity, combined with other signatures that are currently available, provides the first direct evidence that large twisted flux tubes do merge to create active regions." However, I do agree that the magnetic winding does provide additional useful evidence to complete the picture, and I strongly encourage the authors to refine their methodology.

We would like to thank the referee again for his/her comments. We have made substantial edits to the paper based on these comments which help clarify the paper's findings significantly. We hope they are to the referee's satisfaction.

Yours sincerely,

D. MacTaggart, C. Prior, B. Raphaldini, P. Romano and S.L. Guglielmino

(a)

(b)

Figure 1: The ‘serpentine’ emergence of a twisted flux tube with two (cancelling) mixed helicity perturbations. (a) shows the total winding accumulation. (b) shows an isosurface of B_y just before the first partial emergence indicated in (a). The transparent slice is the photosphere.

Figure 2: A magnetogram of B_z at $t = 1800$ s.

(a)

(b)

Figure 3: The winding (a) and helicity (b) accumulations with different magnetic field thresholds.

(a)

(b)

Figure 4: The positions, at $t = 45$ hours of (a) strong transverse field ($|\mathbf{B}| > 100\text{G}$ and $|B_z| < 40\text{G}$) and (b) the winding distribution at this time.

(a)

(b)

Figure 5: The positions, at $t = 45$ hours of (a) strong vertical field ($|\mathbf{B}| > 100\text{G}$ and $|B_z| > 90\text{G}$) and (b) the helicity distribution at this time.

Figure 6: Plots of (a) $\langle B_z^2 \rangle$ and (b) the ratio $H/(L\langle B_z^2 \rangle)$. Gaps and regions of zero flux are due to the period of missing data.

REVIEWER COMMENTS

Reviewer #1 (Remarks to the Author):

The revisions are satisfactory for publication.

Reviewer #2 (Remarks to the Author):

I thank the authors for their detailed answer to my concerns and the changes introduced. I am happy with the revised version of the manuscript and recommend its acceptance.

Reviewer #3 (Remarks to the Author):

I would like to thank the authors for a very thoughtful revision of the paper and the substantial additional detail they provided in their reply to the reviewers.

The authors do provide now a sufficient motivation for their study and the conclusions are fully in accordance with the evidence presented. I recommend publication.

I have 2 minor comments:

line 13, 14 : "there has not yet been any direct observational evidence that can clearly identify the emergence of large pretwisted magnetic flux tubes"

Strictly speaking the observations have been there (the paper re-analyses existing observations), but what was missing was the methodology to extract the relevant information and that is provided in this paper. The authors may want to revise this statement to clarify that.

line 236-238: "two-stage" emergence. I'm under the impression that the two-stage emergence was primarily found in flux emergence simulations that did not include convection. There the flux-tube expand and forms a smooth "pancake" in the photosphere and only emerges further after secondary instabilities driven by magnetic gradients lead to further emergence. Is that behavior also seen in simulations with convection where the flux tube arrives in the photosphere in form of many small patches and loops that can quickly drain mass and continue to emerge?

Referee 1

The revisions are satisfactory for publication.

We thank you for this recommendation and for helping us to improve the paper substantially.

Yours sincerely,

D. MacTaggart, C. Prior, B. Raphaldini, P. Romano and S.L. Guglielmino

Referee 2

I thank the authors for their detailed answer to my concerns and the changes introduced. I am happy with the revised version of the manuscript and recommend its acceptance.

We thank you for this recommendation and for helping us to improve the paper substantially.

Yours sincerely,

D. MacTaggart, C. Prior, B. Raphaldini, P. Romano and S.L. Guglielmino

Referee 3

I would like to thank the authors for a very thoughtful revision of the paper and the substantial additional detail they provided in their reply to the reviewers.

The authors do provide now a sufficient motivation for their study and the conclusions are fully in accordance with the evidence presented. I recommend publication.

We thank you for this recommendation and for helping us to improve the paper substantially.

I have 2 minor comments:

line 13, 14 : “there has not yet been any direct observational evidence that can clearly identify the emergence of large pretwisted magnetic flux tubes”

Strictly speaking the observations have been there (the paper re-analyses existing observations), but what was missing was the methodology to extract the relevant information and that is provided in this paper. The authors may want to revise this statement to clarify that.

Yes, we agree and have changed the quoted line to the following: “there has not yet been a methodology that can clearly and directly identify the emergence of large pre-twisted magnetic flux tubes.” This edit can be found on lines 13-15.

line 236-238: “two-stage” emergence. I’m under the impression that the two-stage emergence was primarily found in flux emergence simulations that did not include convection. There the flux-tube expand and forms a smooth “pancake” in the photosphere and only emerges further after secondary instabilities driven by magnetic gradients lead to further emergence. Is that behavior also seen in simulations with convection where the flux tube arrives in the photosphere in form of many small patches and loops that can quickly drain mass and continue to emerge?

Yes, the “two-stage” emergence is a feature that can be found in flux emergence simulations with convection. As an example, we reproduce Figure 17 from MacTaggart and Prior 2021, GAFD, **115**, 85. This is a simulation of a twisted flux tube creating a bipolar active region. Panel (a) shows the tube shortly after it has reached the photosphere (indicated by the slice). Panel (b) shows field lines of the tube from the side, at this time. Later, the flux tube has expanded laterally at the photosphere (panel (c)) but has not developed higher up in the atmosphere (panel (d)). At a later time (panels (e) and (f)), the flux tube has now been able to expand into the corona. Notice that the lateral expansion of the tube has not changed significantly from panels (c) to (e) and that the upward expansion happens after the lateral expansion. This is an example of “two-stage” emergence, where there is a build up at the photosphere first before the expansion into the atmosphere. Other examples can also be found in the cited work. We have now cited the GAFD paper explicitly on line 235, together with the citations to review papers that cover mainly non-convective cases, as an example of a convective simulation which shows the “two-stage” emergence clearly.

Figure 1: A reproduction of Figure 17 from MacTaggart and Prior 2021, *GAFD*, **115**, 85, which shows the emergence of a twisted flux tube. See that work for details of the simulation parameters.

Yours sincerely,

D. MacTaggart, C. Prior, B. Raphaldini, P. Romano and S.L. Guglielmino

REVIEWERS' COMMENTS

Reviewer #3 (Remarks to the Author):

I recommend publication of the paper in its current form.

Referee 3

I recommend publication of the paper in its current form.

We thank you for this recommendation and for the time you have dedicated to helping us improve this work.

Yours sincerely,

D. MacTaggart, C. Prior, B. Raphaldini, P. Romano and S.L. Guglielmino